# Street Trees for Bicyclists, Pedestrians, and Vehicle Drivers: A Systematic Multimodal Review

**Theodore S. Eisenman** [1,*] **, Alicia F. Coleman** [1] **and Gregory LaBombard** [2]

1   Department of Landscape Architecture and Regional Planning, University of Massachusetts Amherst, Amherst, MA 01003, USA; afcoleman@umass.edu
2   Rhode Island Public Transit Authority, Providence, RI 02907, USA; greg_labombard@comcast.net
*   Correspondence: teisenman@umass.edu

**Abstract:** Multimodal Complete Streets have emerged as a prominent aspiration of urban planning to ensure safe access for all users of streets including pedestrians, bicyclists, motorists, and transit users. Concurrently, municipal leaders are pursuing ambitious tree planting initiatives. These co-arising trends are potentially good news, as trees are important elements of livable cities and Complete Streets. Yet, street trees may have different health and safety benefits and disbenefits for various circulation modes. To advance a multimodal approach to research and practice, we undertook a systematic literature review with goals to (1) identify the scholarly literature addressing links between street trees, human health, and safety for pedestrians, bicyclists, and vehicle drivers; (2) depict the principal disciplines, themes, and conceptual scope of this research; and (3) discuss the implications for urban planning and design practice and research. This review drew upon 13 scholarly databases and yielded 63 relevant articles spanning 15 countries, of which 49 constituted original research. The systematic analysis covers eight research categories. Findings show exponential growth in related scholarship over the past two decades, especially for pedestrians. Journals oriented toward interdisciplinary planning and public health and safety are leading this rise, and benefits far outweigh disbenefits. Yet, there are multimodal tensions especially as it relates to the role of street trees in relationship to drivers and pedestrians. Implications for research and practice are discussed, with an eye towards governance, design, and equity.

**Keywords:** street trees; complete streets; travelscapes; urban greening

## 1. Introduction

Complete Streets are now a prominent aspiration of urban planning, with a goal to "ensure the same rights and safe access for all users of streets, including pedestrians, bicyclists, motorists, and transit users of all ages and abilities" [1]. This is partially predicated on the recognition that people worldwide are gravitating toward cities, and that streets constitute some 80% of the urban public realm [2]. Yet, many urban streets have been designed primarily for motor vehicles, and while global pedestrian mortality from vehicle collisions dropped 28% between 2007–2016, this is still well below the World Health Organization's 50% targeted reduction [3]. As such, municipal leaders and urban designers are promoting street access, safety, and mobility for all users [4].

In the United States, the state of Oregon has been credited with starting the Complete Street movement with its 1971 "bike bill," which mandated that a portion of highway funds be set aside for bike paths [5]. Since then, over 1400 Complete Streets policies have been passed in the United States including 33 by state governments, and 90 percent of them have been enacted in the last decade [6]. Complete Streets represent a shift from transportation planning that privileges motor vehicles toward a multimodal approach where bicyclists, pedestrians, vehicle drivers, and transit users all have a stake. This is important for the most vulnerable people who use streets, including children, people living with disabilities, older adults, and those who cannot afford or do not have access to a car.

More broadly, Complete Streets are understood to improve public health by promoting physical activity through walking and biking, increasing safety for non-vehicle users, and reducing transportation costs and traffic congestion [7]. Complete Streets and related strategies such as "slow streets" and "open streets"–designed to limit through-traffic and function as reclaimed public space–have gained additional traction during the ongoing COVID-19 pandemic [8–11].

Co-arising with contemporary efforts to improve urban streets for all users, there has in recent years been a bloom of interest in urban greening, defined as organized or semi-organized efforts to introduce, conserve, or manage outdoor vegetation in urban areas [12,13]. In many cases, greening includes substantial tree planting. In 2011, London established a goal to increase tree canopy cover from 20 to 25 percent by 2025, equal to 2 million additional trees [14,15]. Several cities in China have million tree planting programs [16], and leaders in Lima, Peru aim to plant 2 million trees between 2019–2022 [17]. Across the United States, municipal leaders have likewise established ambitious canopy cover goals and major tree planting initiatives (TPIs), including programs to plant a million trees [18–21]; New York City met its million-tree planting goal in less than a decade [22]. Importantly, a nationwide survey of U.S. TPIs found that half of trees planted during these campaigns are along streets [12].

This suggests that there may be a noteworthy increase in urban street trees in forthcoming years, and this raises important questions about the relationship between street trees and the health and safety of different street users. This is all the more important as trees are an important component of Complete Streets [6], creating a more pleasing setting for pedestrians and bicyclists, slowing traffic, offering shade, and providing a safety buffer between pedestrians and vehicles. Yet, trees adjacent to streets can also be a potential safety hazard to vehicle drivers, as well as bicyclists and pedestrians due to falling limbs and debris and by bucking sidewalks [13]. These trade-offs are important considerations in the design and management of streetscapes, yet street tree planting and management—depending on national and local context—can often fall upon a hybrid network of public, private, and nonprofit actors [12,23,24]. Even within the public domain, street tree management often crosses administrative and disciplinary jurisdictions including departments of public works, transportation, parks and recreation, and planning [21,25], as well as state and municipal boundaries [26]. This uncertain governance structure assumes added significance when we consider street trees as a form of living green infrastructure that requires substantial human engagement–from design and installation, to establishment, management, and removal [27,28].

These practical challenges have important implications for scholarly research and practice, where different disciplines can study a similar topic on urban trees but cite none of the same literature [29], and arrive at different conclusions about outcomes [30]. These differences reflect and reproduce "disciplinary crosstalk"—poor communication, unconscious misunderstandings, and inconsistent use of terms and literature between disciplines [29–32]. In the case at hand, there has not been much research, and to the best of our knowledge no systematic review, on the relationship between street trees and the health and safety of all roadway circulation modes including pedestrians, bicyclists, and drivers.

In addition to the aforementioned points, there is the following important context for the study at hand. People in the United States spend on average 87% of their time indoors and 6% of their time inside a vehicle [33]—similar findings may apply elsewhere. Of the remaining 7% of time spent outdoors, a significant portion is in all likelihood walking or biking to destinations, because transportation physical activity (e.g., biking/walking to work, school, and shopping, as distinct from recreation activity) is a major contributor to total physical activity [34,35]. This may be especially true in urban settings [36], where the vast majority of people will reside in the foreseeable future [37]. In other words, the places that people move through on a daily basis—also known as "travelscapes"—may be the dominant types of landscape that people routinely experience [38]. Indeed, treating streets

as essential elements of the public realm is foundational to creating livable, sustainable cities [2]. This lends additional urgency to scholarship that addresses the health and safety implications of street trees for all circulation modes.

In response to the aforementioned gaps and opportunities, we conducted a systematic literature review on links between street trees and multimodal safety, with an eye towards implications for Complete Street design in urban centers. Our objectives were to: (1) identify the body of scholarly literature addressing links between street trees, human health, and safety for pedestrians, bicyclists, and vehicle drivers; (2) depict the principal disciplines, themes, and conceptual scope of this research corpus; and (3) discuss the implications for urban planning and design practice and research.

## 2. Materials and Methods

The methodology of this literature review draws upon precedents in urban forestry and landscape scholarship [39–42], as well as guidance for conducting systematic reviews [43,44]. We identified the literature to be reviewed in this study by searching keyword terms in 13 electronic databases between 18 April 2020, and 5 May 2020. Multiple databases were consulted because an initial search in a common database (Web of Science) yielded limited results. We also wanted to capture literature from the most relevant journals in landscape planning, transportation, and urban forestry, which are covered by different scholarly databases. Keyword terms were established based on the particular focus of this study, namely, links between street trees and human health and safety for bicyclists, drivers, and pedestrians. All searches were prefixed with the primary keywords "street trees" AND safety * OR health * followed by discrete searches using each of the following secondary terms: "complete streets", vehicle *, traffic *, pedestrian *, walkability *, walking *, bicyclists *, bicycle *, bicycling *. Note that the asterisk (*) truncation symbol was used to capture root words that have multiple endings.

With a goal to narrow this review to articles focusing specifically on links between street trees and the health and safety of bicyclists, drivers, or pedestrians, two authors screened the titles, abstracts, and content of the initial article pool ($n = 1411$), and excluded papers that did not meet these criteria. This included, for example, studies on street tree anatomy and physiology; disease and pest control; or articles that merely mentioned street trees as a minor element of research on unrelated topics. Similarly, articles were removed if they focused on tree-lined highways since this type of road is rarely accessible to pedestrians and bicyclists. Books and grey literature (e.g., professional reports, popular press, and conference papers) were also excluded. We did not restrict the article search by publication date. The final sample includes articles addressing links between street trees and the health and safety of bicyclists, drivers, or pedestrians ($n = 62$). A flow chart of the literature search, screening, and eligibility process is depicted in Figure 1 (modeled on PRISMA) [42].

This database search for studies addressing links between street trees and human health and safety for bicyclists, drivers, and pedestrians yielded nine literature reviews and four papers on street tree planning and management. These are not conducive to systematic coding as they draw upon literature from multiple sources, so they are reviewed in narrative form in Section 3. All 49 original research articles were, in turn, systematically coded for descriptive characteristics (e.g., publication year, journal title, study location). Each of these studies was also classified based on the type of circulation mode (biking, driving, walking) and the number of circulation modes that it addressed: unimodal studies assess direct links between street trees and one circulation mode; bimodal studies assess direct links between street trees and a combination of two circulation modes; trimodal studies assess direct links between street trees and all three circulation modes. The aforementioned codes were based upon deductive (a priori) categories.

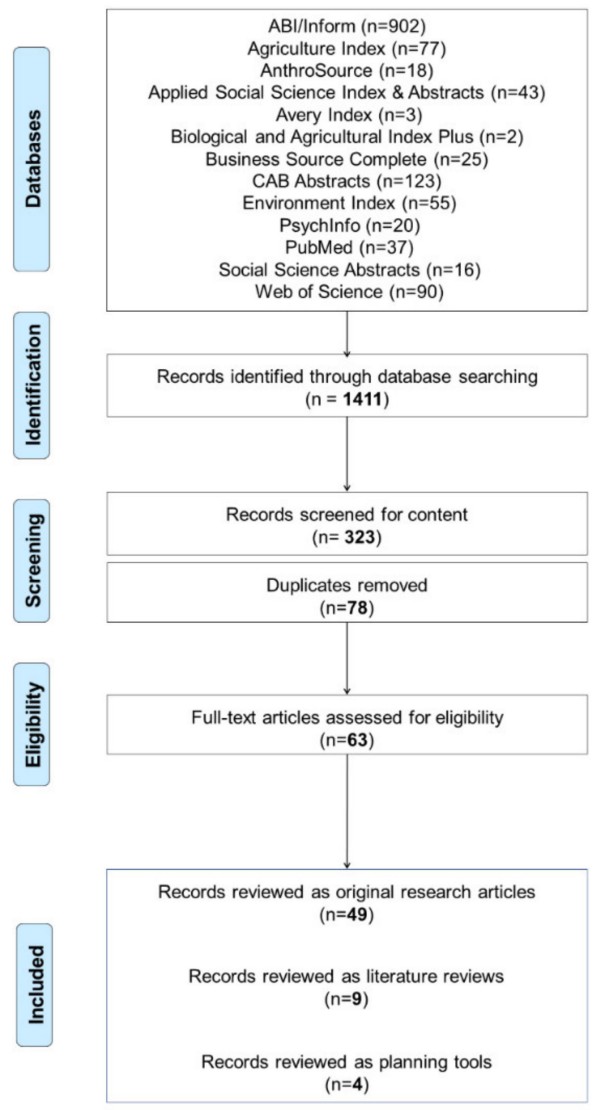

**Figure 1.** Flow chart describing the literature databases searched and the article screening, eligibility, and inclusion process.

To provide greater analytic depth, the 49 original research articles were subsequently coded based upon inductively generated (a posteriori) themes related to the human health and safety *benefits* (e.g., increased physical activity, beautification) or *disbenefits* (e.g., reduced sightlines, decreased perceptions of safety) of street trees. This inductive coding also identified the particular mode(s), as well as the number (uni-, bi, tri-) of modes under investigation in the study. In instances where a benefit or disbenefit did not apply to a particular circulation mode, or where a direct link between street trees and the circulation mode was not explicitly measured (even if it was inferred), the study was coded as unspecified mode. Three broad topics of research fell into this classification: air quality, air temperature, and landscape condition. This is addressed at greater length in the discussion section. To ascertain how different disciplines approach the topic at hand, health/safety benefits and disbenefits were classified by journal discipline. Additionally, we sought to better understand the health/safety benefits and disbenefits of street trees by each circulation mode. The aforementioned combination of deductive and inductive methods increases analytic rigor [45]. To further strengthen reliability, all three study authors participated in the creation and coding of inductive themes to generate a mutually agreed upon classification scheme [46,47].

## 3. Literature Reviews and Planning/Management

### 3.1. Literature Reviews

Van Treese et al. (2017) reviewed research on links between roadway vegetation and drivers' health and risks associated with single-vehicle crashes [48]. They found that in general, roadside flora (including trees) had a positive psychological effect (e.g., reduced stress and frustration) on drivers and was correlated with reduced driving speeds. This was especially true in urban streets with mixed uses and amenities that enhance livability (e.g., planting beds and trees), where several studies showed reduced speed or crashes compared to rural areas. A related review of benefits provided by urban open green spaces also alluded to the capacity of street trees to reduce the speed of vehicles and buffer pedestrians from traffic [49].

These findings support an earlier review of empirical evidence by Ewing and Dumbaugh (2009) addressing the relationship between traffic safety and the built environment, with special attention to how urban planning and design can help or hurt associated outcomes [50]. Contrary to prevailing theory in transportation engineering, this review concluded that the stop-and-go, high-volume traffic environment of dense urban areas appears to be safer than the low-volume settings of suburbs; moreover, the 'less-forgiving' design of urban streets–including narrow lanes, traffic-calming measures, and street trees close to the roadway–seem to improve safety when compared to more conventional roadway designs that emphasize wide lanes and clear zones devoid of such elements. The authors suggest that the reason for this apparent anomaly may be that the less-forgiving but multi-use streetscape designs typical of dense urban areas provide drivers with landscape-based cues that prompt slower speeds and safer driving behavior. In short, landscape setting and context matters. Additionally, this review cited evidence that in urban areas there is safety in numbers: the more bicyclists and pedestrians that share the street, the less per capita incidence of traffic injury.

In a broad synthesis of ecosystem services and disservices of streetscape vegetation, Säumel et al. (2016) offer some noteworthy insights pertaining specifically to street trees [51]. They note that scholarship is biased towards regulating services (e.g., temperature, air quality, $CO_2$ levels, and stormwater) as opposed to cultural or provisioning services. Street trees have at best a minor impact on $CO_2$ sequestration and may even increase net $CO_2$ emissions due to management practices that use fossil fuels. Trees help to reduce streetscape temperatures and manage stormwater. Long-lived trees along streets can, in turn, provide important genetic material (a provisioning service), and trees also provide important cultural services by making streets more attractive and comfortable to be in. However, as street trees age, they present safety hazards, and street trees can also reduce air quality, both of which constitute disservices.

Two review papers focused specifically upon air quality. In recognition of growing interest in stormwater management through vegetated green infrastructure (GI) systems, Shaneyfelt et al. (2017) note that street trees have the potential to improve air quality through deposition of particle pollution onto tree surfaces; however, in relatively narrow streets flanked by buildings–a common urban condition known as 'street canyons'–trees tend to decrease air flow and concentrate air pollutants where people walk, bike, and drive [52]. To minimize this detrimental air quality effect, the authors recommend increasing the space between street trees, pruning to reduce crown size, and selecting low canopy species. In a similar vein, Hewitt et al. (2020) reinforce that GI is often characterized as a solution to urban air pollution, but they note that this is based on conflicting and/or weak evidence; moreover, the impact of GI on air quality depends upon context, with models suggesting that GI can improve urban air quality in some situations, but be ineffective or detrimental in others [53]. In response, the authors propose six types of GI design that can improve air quality-described as GI4AQ—while reinforcing that this will always be a third-order option after reducing emissions and extending the distance between air pollution sources and people. In addition to the aforementioned recommendations by Shaneyfelt et al. (2017), Hewitt et al. (2020) reinforce that planting campaigns

seeking to increase urban tree canopy by 0–10% should select species with low emission of biogenic volatile organic compounds (BVOC), as these gases can create ground-level ozone (a harmful air pollutant) when they interact with oxides of nitrogen in the presence of sunlight.

Mullaney et al. (2015) also summarize the benefits of street trees, prior to addressing the inherent challenges of managing street trees in hardened surfaces and compacted soils [54]. Their review cites a range of environmental benefits including air pollution reduction, stormwater management, $CO_2$ storage, providing shade, and increasing biodiversity and wildlife habitat. However, unlike the aforementioned reviews, this paper does not acknowledge the air quality disservices of street trees. They also cite a range of economic and social benefits of trees such as reduced home heating costs, increased business income, higher property values, increased physical activity, improved social cohesion, and reduced crime, but their review conflates literature focusing on street trees with studies on trees in other landscape settings. Shifting to management challenges and solutions, the authors note that tree root damage of pavement and curbs can be costly, labor-intensive, and present human health risks leading to litigation. In California, for example, 14 cities reported a combined annual trip- and fall-payout of USD1.77 million due to pavement damage from tree roots [55]. To mitigate these risks, Mullaney et al. (2015) propose the use of root barriers, but they note that this can cause tree instability by restricting the growth of root systems in certain directions [54]. They also propose the use of permeable paving and structural soil (designed to maintain roughly 30% porosity and reduce compaction) to increase root access to oxygen and water and deepen root penetration.

Reflecting the complexity and uncertainty of managing ecosystem functions in urban settings where green elements such as street trees often consist of small-scale nodes, Green et al. (2016) advocate for an adaptive and iterative management approach that acknowledges the changing social order of post-industrial cities [56]. The authors note that street trees can generate both services (e.g., reduced stormwater runoff and enhanced aesthetics) and disservices (e.g., infrastructure conflicts and allergens), and street tree management can extend across multiple scales from private households to neighborhood groups to municipal governments. The authors illuminate challenges that can emerge from this decentralized context, including a lack of personal incentive to manage street trees and associated problems of 'free-riding,' where certain parties benefit from other people's engagement and investment in a public good.

Reeve et al. (2015) also address planning and governance themes related to urban flora, and more specifically they advocate for 'biophilic urbanism' [57]. Based on biophilia theory–the proposition that humans possess an evolutionary affection for other forms of life–this builds on the work of Timothy Beatley, who offers direction for how urban planners and designers can increase people's experiences of nature that stimulates positive psychological and physiological responses. In support of this goal, the authors cite a range of biophilic benefits, and those pertaining to street trees include mitigating driver stress, reducing 'traffic incidences' (it is unclear what this means, but we presume the authors are referring to automobile crashes with other vehicles or street users), encouraging active transport, reducing stormwater runoff, and reducing urban heat. Reeve et al. (2015) also cite 'extended infrastructure longevity' as a benefit of street trees, but they do not elaborate on this, which is problematic in light of the aforementioned review by Mullaney et al. (2015), which focuses on mitigating the infrastructural damage caused by street trees. This review closes with a case study of Berlin, Germany, illuminating a range of multi-scalar strategies to increase urban vegetation, such as the Biotope Area Factor (requiring a proportion of "ecologically effective surface area" in redeveloped properties), the Courtyard Greening Program (a 1983 to 1996 project that provided moderate financial support for green roofs and facades), and a recent EUR 2 million street tree planting program initiated despite substantial municipal budget restrictions.

*3.2. Planning & Management*

In an effort to improve microclimatic cooling along different types of urban streets in Orestiada, Greece, Rantzoudi and Georgi (2017) created a classification scheme based on street width and orientation, and categorization of street trees according to height, crown diameter, and distance to adjacent buildings [58]. In addition to providing a more granular approach to designing cooler streetscapes, this paper also notes that street trees separate sidewalks from vehicular traffic, and can also reduce the energy use of adjacent buildings if they are planted on the west and southwest side of building facades.

Noting that most bikeability assessment methods are location- or facility-based, Lin and Wei (2018) developed a new method for assessing the bikeability of a geographically defined area, and they used the Daan District of Taipei, capital city of Taiwan, as a case for testing their method [59]. In this humid subtropical setting where summertime high temperatures average over 90F, street tree shade is considered an important amenity for bikeability. The authors measured the number of street trees per total road length of the area, and along with 24 other variables, assessed the bikeability of zones within the district. Focusing on the use of a public bike-sharing program in Seattle, Washington, Sun et al. (2018) investigated the effects of numerous variables including land use, roadway design, elevation, bus trips, weather, and temporal factors on three-hour long bike pickup [60]. Street trees were identified as a variable that can enhance bikeability, however, street trees were removed from the final model due to problems associated with multicollinearity (when one independent variable is highly correlated with one or more of the other independent variables in a multiple regression equation).

In an effort to develop what the authors describe as a hybrid decision-making approach to roadside plant selection, Guneroglu et al. (2019) established five plant selection criteria and 41 sub-variables based on research literature, then had seven landscape architects rank the sub-variables based on the degree to which they influenced plant selection for roadside settings [61]. Of the five criteria, structural traits (e.g., size, growth, and rooting) ranked highest, followed by economic and ecological variables. It is noteworthy that the traits pertaining most to human health and safety–described by the authors as functional characteristics that include screening of light from oncoming traffic, separating pedestrians from vehicles, improving air quality, and reducing stress–ranked fourth, followed by visual characteristics (e.g., leaf and flower color, texture, and form).

## 4. Results

Descriptive and analytic coding of the original studies that emerged from our search (*n* = 49) is presented below in the same order as the review categories in Table 1. Our study is especially interested in the multimodal dimension of links between street trees and human health and safety, so results are presented with mode type and distribution in mind.

**Table 1.** Descriptions and codes of review categories.

| # | Review Category | Description | Code |
|---|---|---|---|
| 1 | Mode Distribution | Number of circulation modes studied (if applicable) | Text: e.g., unimodal, bimodal, trimodal |
| 2 | Mode Type | Circulation mode(s) studied (if applicable) | Text: e.g., biking, driving, walking |
| 3 | Publication Year | Year of publication | Numeric; e.g., 2009 |
| 4 | Journal Name | Journal of publication | Text: e.g., *Landscape and Urban Planning, Journal of Safety Studies* |
| 5 | Journal Discipline | Disciplinary orientation of study based on the journal's mission statement | Text: e.g., transportation, interdisciplinary planning |

<div align="center">

**Table 1.** *Cont.*

</div>

| # | Review Category | Description | Code |
|---|---|---|---|
| 6 | Study Location | Continent, country, and city where the study took place | Text: e.g., North America, United States, New York, NY |
| 7 | Health and safety benefit | Type of direct human health and safety benefit(s) studied for bicyclists, drivers and pedestrians | Text: decreased crash rate and severity, increased mental health, increased perceived safety, |
| 8 | Health and safety disbenefit | Type of direct health and safety disbenefit(s) studied for bicyclists, drivers and pedestrians | Text: increased crash rate & severity, impaired sightlines, decreased perceived safety, buckling sidewalks |

*4.1. Mode Distribution and Type*

As noted in Figure 2, studies that focus solely on links between street trees and one transit mode–unimodal research–constituted nearly half (47%, *n* = 25) of the sample. Of the unimodal studies, most focused on pedestrians (57%, *n* = 13), while articles on drivers (30%, *n* = 7) and bicyclists (13%, *n* = 3) were a smaller proportion of the total sample. Of the research that focused on two modes of transit (18%, *n* = 9), seven studies (78% of bimodal studies) addressed interactions between street trees and both pedestrians and bicyclists—a combined category that is often described as 'active transportation' [62], 'active travel' [63], or 'non-motorized travel' [64]. One bimodal study addressed links between street trees and bicyclists and vehicle drivers [65], and one study addressed links to pedestrians and drivers [66]. No studies in our sample addressed links between street trees and all three modes (trimodal). A sizable proportion of articles studied the health and safety implications of street trees without narrowing outcomes to a particular type of circulation, classified here as unspecified mode (35%, *n* = 17).

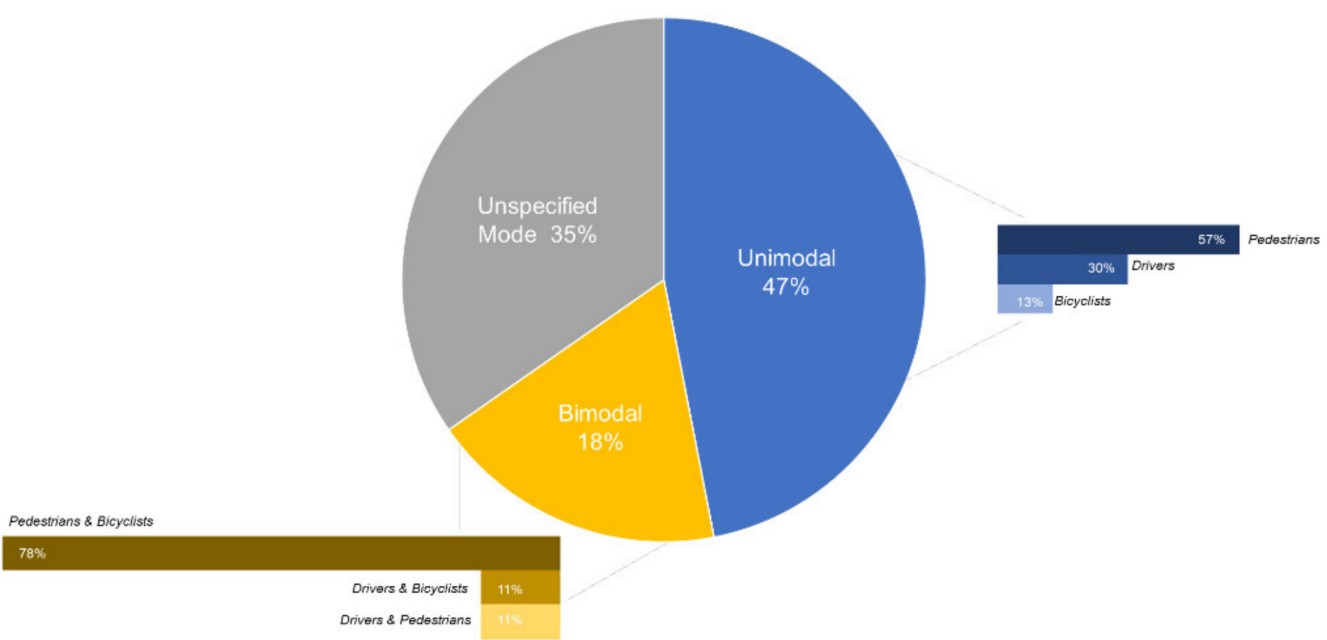

**Figure 2.** Type and distribution of circulation modes.

*4.2. Publication Timeline*

Research on links between street trees and the health and safety of people engaging in different circulation modes has grown substantially since studies emerged in the early 2000s, with a notable increase between 2015–2020 (see Figure 3). The number of studies addressing links between street trees and drivers has been relatively constant, while studies

focusing on pedestrians has grown exponentially in recent years. There has also been a noteworthy increase in studies addressing various dimensions of the streetscape but no specific mode (e.g., air quality, landscape condition, and temperature) in recent years. The most dominant unimodal category (*n* = 14) in our sample focuses on pedestrians. In this sample, research studying the impacts of street trees on bicyclists emerged in 2008, first in combination with pedestrians [67], and then alone [68]. Reflecting the shift towards multimodal transit environments, results from this sample show that bimodal research has consistently increased over the last 15 years.

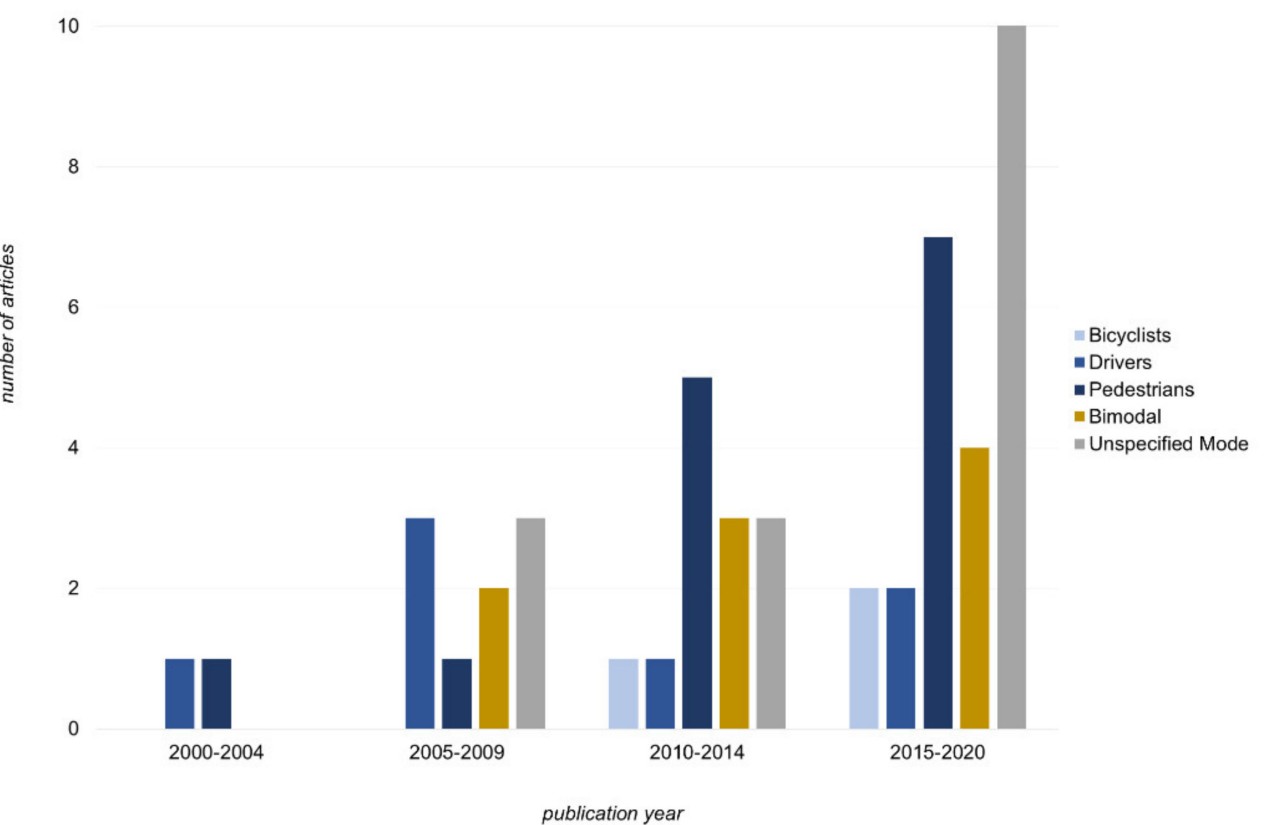

**Figure 3.** Publication timeline by circulation mode.

*4.3. Discipline Distribution*

To broadly scope the disciplinary orientation of scholarship, we reviewed the mission statement of each journal that emerged in our search, and we established the following categories: environmental science, forestry (which encompasses arboriculture–the cultivation, management, and study of individual trees and woody plants), interdisciplinary planning, public health and safety, and transportation. As noted in Figure 4, the greatest number of articles—two-thirds of our sample—emerged from journals in interdisciplinary planning (*n* = 17) followed by public health and safety (*n* = 16). Of the five disciplinary orientations, only interdisciplinary planning and public health and safety journals included studies addressing all mode categories. All disciplinary orientations other than environmental science addressed bimodal circulation, and transportation journals were the only ones to not address pedestrians. Environmental science papers largely consisted of unspecified mode studies (e.g., air quality and temperature).

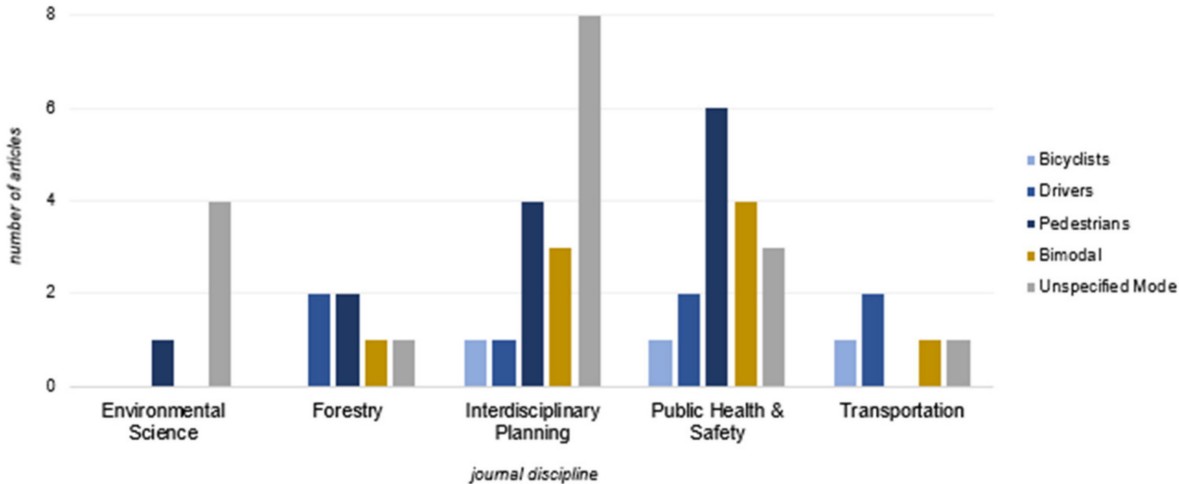

**Figure 4.** Disciplinary distribution by circulation mode studied.

### 4.4. Study Location

Because literature reviews are generally not specific to a place, the 14 studies described in Section 3.1 are not included in this subsection. As noted in Table 2, most original studies on street trees and human health and safety identified in this review were conducted in North America (*n* = 29), and primarily in the United States (*n* = 26). Study areas in Europe (*n* = 10) were roughly distributed between six countries: England (*n* = 2), Germany (*n* = 2), Italy (*n* = 2), Netherlands (*n* = 1), Romania (*n* = 1), and Spain (*n* = 2). Similarly, research in Asia was conducted in five different countries: China (*n* = 2), India (*n* = 1), Japan (*n* = 1), South Korea (*n* = 1), and Syria (*n* = 1). Regions with little or no representation in this sample include Africa, Oceania (*n* = 1), Southeast Asia, the Middle East, and South America (*n* = 1).

**Table 2.** Geographic distribution of original studies. * An original study conducted as a laboratory simulation is not included here because it is not place-specific.

| Continent | Country | Total # Unique Cities | Total # Articles * |
|---|---|---|---|
| North America | Canada, United States | 17 | 29 |
| Europe | England, Germany, Italy, Netherlands, Romania, Spain | 10 | 10 |
| Asia | China, India, Japan, South Korea, Syria | 6 | 7 |
| Oceania | Australia | 1 | 1 |
| South America | Brazil | 1 | 1 |

### 4.5. Health/Safety Benefits and Disbenefits of Street Trees by Journal Discipline

Figure 5 shows that interdisciplinary planning and public health and safety journals account for the most findings of benefits, followed by forestry, transportation, and environmental science. Increased physical activity is the most commonly cited benefit of street trees spanning most journal disciplines, but this type of benefit did not emerge in environmental science and forestry. The most frequently studied benefit across journal disciplines was temperature and thermal comfort, followed by landscape condition/visual preference. The most commonly identified disbenefit of street trees—primarily in public health and safety journals—was crash rate and severity. The second most common disbenefit was air quality, distributed across environmental science and public health and safety journals.

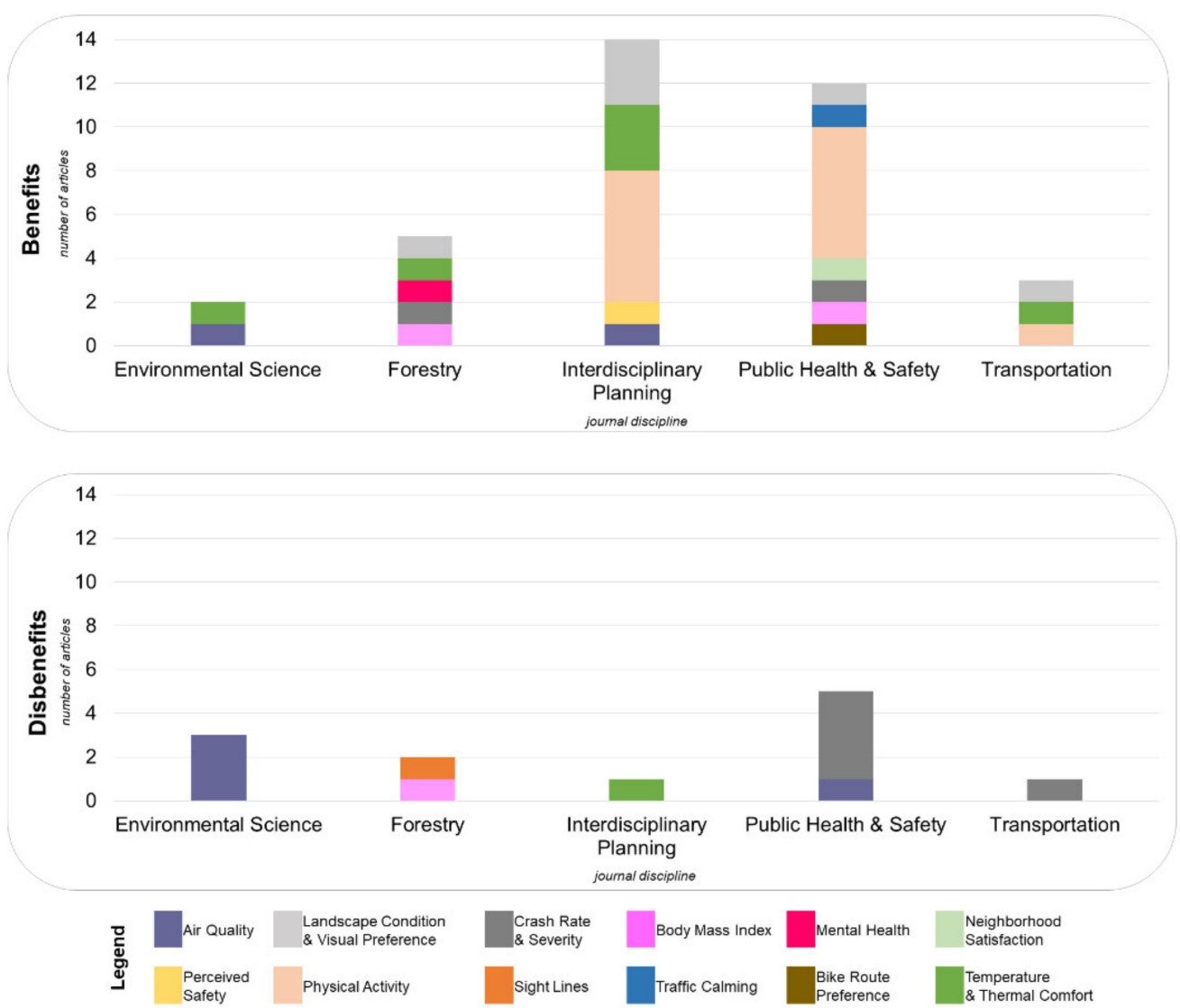

**Figure 5.** Number of findings of health/safety benefits and disbenefits of street trees by journal discipline.

### 4.6. Health/Safety Benefits and Disbenefits of Street Trees by Circulation Mode

Figure 6 depicts the number (size of circle) and type (wedges in circle) of health and safety benefits and disbenefits across circulation modes. This figure separates the bimodal studies into separate benefits or disbenefits by respective mode, thus, total counts are larger here than in previously reported results. As illustrated in Figure 6, there are almost three times more human health and safety benefits ($n = 44$) than disbenefits ($n = 15$) that emerged in our search. Pedestrians accounted for the most total benefits of street trees ($n = 20$), and they also accounted for the greatest variation of benefits ($n = 8$), led by physical activity and followed by body mass index and landscape condition and visual preference. Temperature and thermal comfort were the second most identified benefit across modes and unspecified modes ($n = 6$), although this was not identified as a benefit for drivers per se. While accounting for fewer total studies, physical activity and temperature/thermal comfort had roughly the same distribution amongst bicyclists as pedestrians. Four types of benefits–air quality, landscape condition/visual preference, physical activity, and temperature/thermal comfort–were classified in the unspecified mode category ($n = 11$), as their empirical measures apply to the street corridor writ large and do not demonstrate a direct etiological pathway to a particular mode user (e.g., physical activity may be derived from either or both cycling or walking). The most commonly identified disbenefit of street trees was crash

rate and severity, and drivers accounted for five of these six findings. This was also the only type of street tree disbenefit identified for all three circulation modes. In the unspecified mode category applied to the whole street corridor, air quality was the most commonly identified disbenefit.

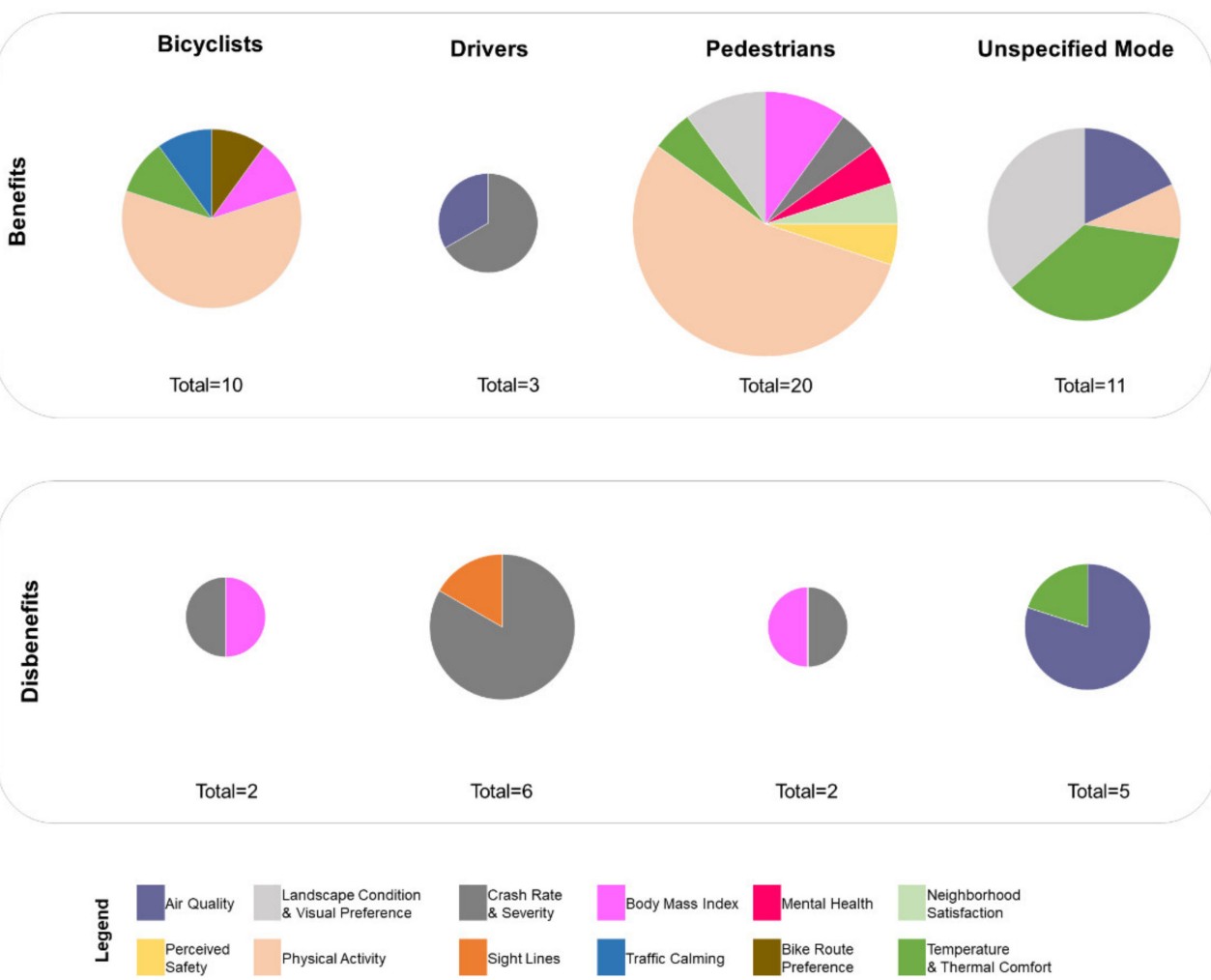

**Figure 6.** Number of findings of health/safety benefits and disbenefits of street trees by circulation mode proportional to category.

## 5. Discussion

An underlying premise of this study is that a multimodal understanding of street trees will help to guide the planning and design of Complete Streets in cities. It is therefore notable that no study emerging from our search addressed the health and safety implications of street trees for all three circulation modes: bicyclists, pedestrians, and vehicle drivers (see Figure 2). However, this is not entirely surprising. From a methodological standpoint, assessing the health and safety outcomes of three circulation modes is more complicated than addressing two or one. Moreover, disciplinary orientation tends to direct the scholarly gaze toward certain objects of study while discounting others [69,70]. Planning and public health/safety journals accounted for twice as many studies on human health and safety links with street trees than environmental science, forestry, and transportation combined (see Figure 4).

This last point is noteworthy. Prior to conducting this study, we would have guessed that many if not most studies in this review would have originated in forestry, as street trees are the common denominator (i.e., the independent variable). Yet, our review spanning

13 scholarly databases only yielded six original studies published in forestry journals that address links between street trees and the human health and safety of bicyclists, pedestrians, or vehicle drivers. This suggests that the tree itself is the likely focus of researchers who publish in these journals, while other scholarly orientations such as planning, public health and safety, and to a lesser extent transportation, undertake research that addresses more directly associated human health and safety outcomes. Most studies of bimodal outcomes were also published in planning and public health/safety (see Figure 4), suggesting that these disciplines play an important role in the scholarship at hand.

Of the nine bimodal studies that emerged in this review, most (*n* = 7) addressed links between street trees and bicyclists and pedestrians, both of which constitute active forms of travel/transportation. This is likely due to the increased focus over the past two decades on the impact that the built environment may have on physical activity and related health problems such as obesity and diabetes. Between 2001 and 2013, for example, the Robert Wood Johnson Foundation committed USD 32.9 million to support 260 original research grants intended to increase physical activity through its Active Living Research program [71]. An early example of this research is Ewing (2002) [72], who found that an impediment to context-sensitive and pedestrian friendly main street design in the U.S. is the reluctance of state departments of transportation to assume management responsibilities for street trees and landscape medians. Challenges related to the design and governance of walkable downtown streets (aka "high streets") have also been noted in England [73]. This is important because the proportion of street frontage covered by tree canopy has been strongly correlated with people's preferences when assessing "main streets"–urban street corridors that are not just channels for vehicular movement but places in their own right, with mixed used functions that accommodate pedestrians and bicyclists [74]. In subsequent and related scholarship, survey respondents in Washington State identified micro-scale design elements such as street trees, benches, and lighting as important for promoting biking and walking [67]. Likewise, interviewees in San Francisco, CA identified street trees as a way to increase enjoyment and safety/security for bicyclists and pedestrians, but the study never explained how street trees contribute to the latter [75]. In New York City, Lovasi et al. (2013) found that in low-poverty areas increased street tree density was marginally significant for increased active transportation (bicycling and walking), yet, no such benefit was observed in high-poverty neighborhoods [62]. These kinds of mixed findings are also reflected in London, Ontario, where two studies found that the likelihood of walking or biking to school was positively associated with more street trees, yet, no such correlation was identified for the trip from school to home [63,64]. That is because the morning commute to school corresponds with parents driving to work, so there are more mode options for students, but getting driven home in the afternoon is often not an option because schools in this place generally close around 3:00 p.m. before the workday ends; so, the environment plays a larger role on the trip to school (K. Larsen, personal communication, 10 May 2021).

Shifting to bimodal studies that include motorists, Kim (2019) found that in Los Angeles, CA street trees can increase the safety of elderly pedestrians at intersections by protecting them from collisions with vehicles, and by providing both pedestrians and drivers with a clear definition of roadways and sidewalks [66]. Meanwhile, street trees can also block the view of both drivers and pedestrians. Nevertheless, they recommend planting street trees in senior slow zones and along safe routes for seniors. In Seattle, WA, Chen (2015) did not find that street tree density significantly reduced crash frequency between bicyclists and vehicles [65].

For bicyclists, street trees have been shown to positively contribute to biking route preferences and the overall livability of a city [76], although this line of research is in its infancy and more research is needed to extend generalizability. Street trees can encourage bicycling as a comfortable, safe mode of active transportation [68] and increase the appeal of a specific route; these benefits are accomplished by providing shade and creating perceptible beauty along roadways [68,77]. In this small area of research, there has been little

exploration of the difference between utility bicyclists (who rely on bikes as a prominent mode of transportation) and leisure cyclists (who primarily cycle for recreation). This distinction is acknowledged in Nawrath et al. (2019) [76], whose leisure-cyclist respondents said they value green streets and would change their biking route to be near more greenery; utility and regular bicyclists, on the other hand, valued the street settings without vegetation higher than leisure and all-rounder cyclists. This led the researchers to conclude that regular utility cyclists have a stronger focus on the efficiency of routes–not street trees. In other words, urban street greening may play a more important role for leisure cycling. Another important takeaway is the negative effect of street trees on bicycling infrastructure (an ecosystem disservice) [13]. Issues such as branches obstructing line of sight, overgrowth impeding a designated bike path, roots buckling roadway surfaces, and defoliated leaves creating slippery surfaces can, if not managed adequately, can compromise cycling as a safe mode of transit [76].

Of the studies linking street trees to the health/safety of vehicle drivers, more original research noted disbenefits (*n* = 4) than benefits (*n* = 1), although crash incidence and severity is the most pervasive outcome measure held in common. Street trees can reduce crash incidence and severity by creating an arching canopy that calms traffic by narrowing the street corridor, improving the aesthetic quality of roadsides and thereby reducing drivers' physiological stress response (when compared to scenes with buildings), and relaying positive cues about community character [78]. However, the attributes that make street trees an amenity (e.g., large branches, dense foliation, wide trunks) do not conform to standard roadside "clear zones" and may decrease the health/safety of vehicle drivers by decreasing visibility and obstructing drivers' sightlines [79,80]. These disbenefits may not hold true across all roadways, as noted by Ewing and Dumbaugh (2009) [50]. The authors counter that the assumptions and design standards of "clear zones" may apply differently between high-speed rural roads and low-speed urban streets and supporting research (from their review) has demonstrated that street trees, specifically, can moderate crash incidence and severity differently across different types of roadways. This point is further emphasized by a more recent study found in our present review, where Chen et al. (2016) concluded that improper planting and maintenance—not merely the presence of trees in roadway medians and intersections–increased crash incidence and severity; the authors recommended that landscape maintenance should be a normalized form of "engineering technology" to reduce vehicle-related injuries or fatal crashes [81].

Of the three circulation modes addressed in this review, pedestrians may be the greatest beneficiary of street trees (see Figure 6). Several studies show positive links between the presence of street trees and increased walking [82–84]. Related studies also show beneficial associations between street trees and reduced body mass index (BMI) or prevalence of obesity [85–88], outcomes that were in all likelihood due primarily to increased walking. He et al. (2020) found that street greenery including trees was positively associated with older adults in Wuhan, China achieving >300 min of pedestrian-centered physical activity per week, but park area was not [89]. This reinforces the importance of streets as travelscapes and essential elements of the urban public realm. Street trees may also benefit pedestrians by reducing vehicle speeds and buffering pedestrians from traffic [49]. The elderly, in turn, have been shown to be less likely to identify as physically disabled and disabled from going outside the home if their neighborhood has more street trees and intersections [90]. In addition to increasing physical activity, street trees are associated with reduced psychological tension, fatigue, confusion, and anxiety for both men and women [91], as well as improved neighborhood social capital, defined as social resources in community networks that are relevant to the formation of trust, social norms, reciprocity, and mutual support [92]. This is noteworthy because social capital—and related ideas such as social cohesion and connectedness—is a significant determinant of health and well-being [93–95]. The aforementioned studies showing benefits to pedestrians may, in turn, be associated with a preference for tree-lined streetscapes [83], although the extent of such preferences can depend on cultural context. In Sapporo, Japan, one study found that

people preferred sidewalk planting beds of flowers without trees over similar planting beds with trees [96]. Street trees can also function as disservices for pedestrians. In our review, this includes blocking the views of pedestrians and drivers which could increase crashes between the two [66], and increased associations with obesity in areas with large parcels and distance between destinations [88]. In the latter case, however, increased obesity is in all likelihood due to lifestyles associated with low-density suburban form as opposed to the trees themselves.

Our review identified 17 studies that addressed various conditions of the streetscape environment, but because these studies did not necessarily assess etiological outcomes pertaining to a particular type of street circulation, they were classified as unspecified mode (see Figure 2). The most commonly assessed variable in this category is air quality, and these studies show mixed findings. On the one hand, street trees can improve air quality through gravity-based deposition of air pollutants on leaves and branches [97,98]. On the other hand, street trees can decrease air flow and dispersion of air pollutants, and while a higher volume of vegetation increases pollution removal via deposition it also weakens air pollution dispersion [97]. One study, for example, found that planting new street trees had a negligible effect on the average pollutant concentration of a whole neighborhood in Pamplona, Spain; yet, in the street corridor itself, air pollution concentrations increased 12% when densely planted with trees [99]. Trees also emit pollen and biogenic volatile organic compounds (BVOCs)—which create ground-level ozone (a major pollutant) when they interact with oxides of nitrogen in the presence of sunlight–and both of these naturally occurring emissions of trees can impair respiratory health. However, this depends on the particular place. In Sydney, Australia, Sercombe et al. (2011) found that people did not have strong allergic reactions to pollen from London Plane Tree (*Platanus*) [100], even though this commonly planted street tree is generally characterized as highly allergenic [101,102]. The effect of street trees upon air quality is also informed by the width of the street, the density and height of adjacent buildings, tree species, planting density, wind direction, air velocity, and regional land cover and vegetation composition. A full accounting of these interacting variables is beyond the scope of this study (e.g., [30,103]), but for purposes of the review at hand, it is important to distinguish between multimodal, mixed use streets in urban centers and limited access highways (or "open roads") that serve primarily or exclusively motor vehicles and are not intended for pedestrians and cyclists. Additionally, here, the scholarly literature on urban trees and air quality is consistent with the findings of studies that emerged in this review: along mixed-use streets in urban centers—sometimes described as street canyons where building height to street width 'aspect ratio' is >0.50—trees tend to increase air pollution concentrations where people walk, bike, and drive by reducing air flow and concentrating pollutants [97,100,104]. However, adjacent to open or exposed roads dedicated to high volume motor vehicle traffic, trees and vegetation can reduce human exposure to air pollution depending on a range of factors [105,106]. Most importantly, the solution to urban air pollution is not to plant more trees, but to reduce and eliminate air pollution emissions at their source [107–109].

Temperature regulation is a second environmental mechanism whereby trees can impact human health and safety for unspecified mode users, and most studies in our review that addressed this topic showed beneficial cooling effects. However, associated outcomes depend on a range of factors including underlying urban form, time of measurement, and distinctions between surface and air temperature. In settings with relatively low building H/W ratios < 0.50 in Damascus, Syria (streets in the dense urban core were not included because they are too narrow to accommodate trees), the surface temperature of tree-lined streets was up to 16.8 °C cooler than streets without trees [110]. In a somewhat similar urban density in Utrecht, Netherlands (average H/W ratio of 0.48), 10% tree cover lowered mean radiant temperature about 1 °C; however, when asked to reflect upon their direct experience of thermal comfort, interviewees indicated that streets with trees and front gardens were slightly more comfortable but results were not statistically significant [111]. In a study of commercial streets in Seoul, Korea, street trees were identified as the most

effective strategy for reducing surface temperatures [112]; yet, in a counterintuitive finding, road surface temperatures on streets with trees were slightly warmer at night. The authors attribute this to street trees retaining heat generated during the day and reradiating this heat at night. In a low-density suburban setting, Kim et al. (2018) found a roughly 1–2 °C reduction in air temperature along different types of sidewalks that had trees compared to those that did not [113]. Additionally, along open roads in Bangalore, India, tree-lined segments had a maximum air temperature of 34.2 °C, while road segments without trees had maximum ambient temperature of 38.3 °C [106].

A third environmental aspect of street trees that can indirectly impact various modes of circulation, is enhancement of the physical landscape corridor itself. In related studies on main streets (U.S.) and high streets (U.K.), trees were identified as important elements that enhance street character and the visual preference of these mixed-used, multimodal corridors [73,74]. Along two high streets in London, for example, landscape improvements including new crossings, changed road alignment, bicycle parking, footway widening and re-paving, and new street trees yielded a 7% and 30% increase in pedestrians and cyclists, respectively [73]. Improving the condition of streetscapes in neighborhoods with low socioeconomic status may be an especially important strategy to reduce socioeconomic disparities in physical activity among urban residents; this includes planting trees along streets, increasing the frequency of trash pickup and other sanitation services, improving community policing, implementing traffic-calming measures, and promoting economic development in poor neighborhoods [114]. Street trees may also increase bicycling and walking in neighborhoods with varying types of spatial form; however, people's underlying preference for neighborhood form significantly influences their travel behavior [115].

### 5.1. Implications for Practice and Research

This systematic review reveals noteworthy themes to maximize the health and safety benefits of street trees for multiple circulation mode users while minimizing disbenefits–a process that requires weighing various tradeoffs [13]. As noted by studies included in this review and others, the management and governance of urban street trees is often unclear [73,116], and may be perceived as a reluctant undertaking in public works and transportation departments. Lack of clear authority can be exacerbated during urban tree planting initiatives (TPIs), which are often spearheaded by actors who may lack short- and long-term management authority [12,26]. Professional arborists, for example, spend much of their time managing trees to reduce disservices and risks [13,27,117], and vegetation in urban settings requires ongoing care in order to ensure socio-environmental benefits [118]. These are timely considerations as TPIs have become an increasingly common municipal practice and roughly half of trees planted during TPIs are along streets. Indeed, TPIs may require new forms of governance that account for a range of stakeholder attitudes and expertise, and foster greater institutional capacity than are the norm in municipal management of landscapes and traditional grey infrastructure [19,25]. Municipal leaders across administrative units and related nonprofit actors are encouraged to increase cross-disciplinary communication about the goals and implementation of street tree planting and management. This participatory goal setting and management process should include local residents, especially in historically disadvantaged communities [119,120]. The growing adoption of urban forest management plans is one vehicle for doing this [121,122].

Complete Street initiatives are another opportunity to increase coordination of street tree planting and management. This is especially timely in the United States, where a legislative bill—The Complete Streets Act of 2021—was recently introduced in Congress [123]. This proposed legislation responds, in part, to alarming data that the number of people struck by motor vehicles and killed each year in the United States has increased 45% between 2010 and 2019—even though walking as a share of total trips stayed steady—and 2018 and 2019 saw the highest numbers of pedestrian deaths since 1990. Older adults, people of color, and low-income communities are disproportionately represented in fatal

crashes involving pedestrians, even after controlling for differences in population size and walking rates [124]. In light of the current-day discourse on social inequity, this elevates the importance of creating safer urban streets especially in marginalized communities. Additionally, the 2020–2021 COVID-19 pandemic has cultivated new interest in reducing auto-dependence and elevating streetscapes as essential elements of the urban public realm [125,126]. As municipalities explore new ways to retrofit streets to make them more accessible to pedestrians, street trees could figure prominently.

This will require context-sensitive design that accounts for the physical form, materiality, and social dynamics of a given place. As noted by Dumbaugh (2005), different disciplines and countries have distinct approaches to this. In many European cities, designers start with an "environmental reference speed" and use that to guide both posting of speed limits–which tend not to exceed 50 km (31 mph) in urban settings with adjacent roadside development and pedestrian activity–and incorporation of design features such as trees that reinforce the intended reference speed [127]. In the United States, by contrast, road design is advanced primarily by traffic engineers through the *American Association of State Highway and Transportation Roadside Design Guide* or "Green Book" [128], which emphasizes the importance of "clear zones" to avoid vehicle crashes with fixed objects such as trees. Original guidelines recommended that any tree with a mature trunk diameter greater than 4 inches should be planted 30 feet (9.1 m) from the road edge [129], but this is clearly not possible in most urban areas with mixed-use streets. The updated 2011 guidebook recognizes the challenge of these constraints but does not specify offsets or placement guidelines for street trees, leaving the planting location of street trees at the discretion of the attending engineer. Subsequently, the National Association of City Transportation Officials (NACTO) published *The Urban Street Design Guide* [130], which outlines spacing recommendations based on street tree species, property lines, and other roadside features. Predicated on a more urban and multimodal logic, this guidebook only encourages street tree removal in extreme cases, and it recommends that street trees be planted no closer than 5 feet from an intersection or stop sign, recessed 3 feet from curbs, and aligned with the corner of adjacent buildings. Street trees are promoted by the American Society for Landscape Architecture and the Congress for New Urbanism as essential elements of streetscape design [131,132]; street trees also figure prominently in an international survey and analysis of great city streets [133]. Importantly, urban streets that were built primarily for motor vehicles can be redesigned with narrower lanes and street trees closer to the roadway, and in so doing reduce vehicle speeds and crashes between drivers and pedestrians and bicyclists [50].

While aspects of the preceding discussion on U.S. policy and practice may be applicable elsewhere, another consideration that emerges from this study is a need for greater international scholarship (see Table 2). The density and distribution of street trees can, for example, differ substantially between cities in different climates and cultures, and underlying urban form may be an important consideration [134]. Additional considerations include selection of tree species with low allergenicity and BVOCs, and greater scholarly attention to the mental and social benefits of street trees. Over the past three decades, a sizable body of scholarship has consistently shown that people derive a range of psychological benefits from viewing or moving through vegetated landscapes in urban settings [135,136], and stress reduction may be a prominent mechanism explaining associated human health benefits [137]. In an urbanizing world where the vast majority of people will soon be living in cities, and travelscapes are a prominent means by which people interact with outdoor environments on a regular basis, tree-lined streets could be a powerful way to counteract the stresses of urban living [138,139]. These experiential benefits of trees will likely accrue to all mode users of streets including bicyclists, drivers, and pedestrians, suggesting that street trees could be a powerful tool for enhancing the livability of 21st century cities.

*5.2. Limitations*

Parsing research into discrete categories and classes can be a subjective and reductionistic act, both of which have inherent problems. Subjective classification is based on the assessors' interpretation, which can introduce unconscious bias and error [140]. Reductionism can, in turn, oversimplify complex relationships [141]. With these caveats in mind, the categorization and classification undertaken in this study can be helpful when seeking to advance understanding of complex relationships.

## 6. Conclusions

Predicated on rising interest in urban tree planting initiatives (TPIs) and multimodal Complete Streets in cities around the world, we undertook a systematic review of scholarly research addressing links between street trees and the health and safety benefits/disbenefits of bicyclists, pedestrians, and vehicle drivers. This body of scholarship has grown substantially in recent years, but it is dominated by research based in the United States and could benefit from a more international scope. Nearly half of all studies focused on just one circulation mode and none addressed all three modes, suggesting a need for more multimodal scholarship. Of five disciplinary orientations, interdisciplinary planning and public health/safety journals accounted for two-thirds of original studies, suggesting that these fields play an important role in the topic at hand.

Nearly three times more human health and safety benefits than disbenefits emerged in this review, and pedestrians accounted for the most benefits of street trees; they also accounted for the greatest variation of benefits, with increased physical activity constituting the majority of these benefits. Improved temperature/thermal comfort and landscape condition/visual preference were the next most-cited benefits, both of which may enhance all circulation modes. Most bimodal studies addressed links between street trees and bicyclists and pedestrians, both of which constitute active travel/transportation—a key element of Complete Streets. Yet, research on links between street trees and cyclists is fairly nascent and represents an opportunity for further inquiry.

The most commonly identified disbenefit of street trees was vehicle crash rate and severity, followed by reduced air quality in narrow urban street corridors. Importantly, the relationship between street trees and vehicle crashes with pedestrians depends on adjacent landscape context; along low-speed, mixed-use streets in urban centers that account for pedestrian use, trees may reduce crash incidence and severity by calming traffic and clearly demarcating sidewalks from roads. However, street trees need to be managed to avoid blocked views, buckled paving, and surface litter, all of which can create safety risks for cyclists and pedestrians. This requires effective management and communication amongst municipal officials, especially as the governance of street trees and associated streetscape vegetation may not always be clear and can differ between cities. As municipalities pursue TPIs and Complete Street programs, these initiatives present an opportunity to strengthen the institutional capacity for short- and long-term management of urban street trees.

**Author Contributions:** Conceptualization, T.S.E.; methodology, T.S.E., A.F.C., G.L.; software, A.F.C., G.L., T.S.E.; validation, T.S.E., A.F.C., G.L.; formal analysis categories 1,2,7,8, T.S.E., A.F.C., G.L., categories 3,4,5,6 A.F.C., G.L., T.S.E.; data curation, A.F.C., G.L., T.S.E.; writing—original draft preparation, T.S.E., A.F.C., G.L.; writing—review and editing, T.S.E., A.F.C., G.L.; visualization, A.F.C., G.L., T.S.E.; supervision, T.S.E., A.F.C.; project administration, T.S.E., A.F.C., G.L.; funding acquisition, T.S.E., A.F.C. All authors have read and agreed to the published version of the manuscript.

**Funding:** This research was made possible by funding from the USDA National Institute of Food and Agriculture, Massachusetts Agricultural Experiment Station (#MAS00037), and the National Science Foundation award CNH-1924288.

**Institutional Review Board Statement:** Not applicable.

**Informed Consent Statement:** Not applicable.

**Data Availability Statement:** Coding data available upon request to corresponding author.

**Acknowledgments:** We extend sincere thanks to Madeleine Charney, Research Services Librarian, and Jennifer Friedman, Head of Research Services, at the W.E.B. DuBois Library at University of Massachusetts Amherst.

**Conflicts of Interest:** The authors declare no conflict of interest and the funders had no role in the design of the study; in the collection, analyses, or interpretation of data; in the writing of the manuscript, or in the decision to publish the results.

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
