# Peer review of "Street Trees for Bicyclists, Pedestrians, and Vehicle Drivers: A Systematic Multimodal Review"

_urbansci, doi:10.3390/urbansci5030056_

Round 1

Reviewer 1 Report

Dear Authors,

first of all, congratulations on your specific street trees review. Going through the text I really appreciate a very good text structure and the fact that your article is well understandable. Would like to appreciate a high number of adequate references.

There is just one recommendation from my side - to make clear the object of your research in the abstract. And, also, I would like to suggest to put down there a number as well as the name of countries where the original studies have been taken. In this respect the abstract is a bit general.....

Specific comments:

Abstract:

Lines 17 – 23. Please insert the general objective of your review and a very short summary of your main findings. Furthermore, please indicate, very shortly, the main conclusions.

Conclusions:

Please indicate very clearly how can main urban planning stakeholders benefit from your review. Possibly, how can urban designers and landscape architects benefit from your review.

Reviewer 2 Report

The manuscript covers interesting topic on "Complete Streets" for street users including pedestrians and others.

  1. Need more justification for this study liking current problems with streets most of the world.
  2. Please insert more literature review on street trees' function on human health to pedestrians and public.
